# Effects of Boron Content on Microstructure and Wear Properties of FeCoCrNiB$_x$ High-Entropy Alloy Coating by Laser Cladding

**Dezheng Liu [1], Jing Zhao [1,*], Yan Li [1], Wenli Zhu [1] and Liangxu Lin [2,*]**

[1] Hubei Key Laboratory of Power System Design and Test for Electrical Vehicle, Hubei University of Arts and Science, Xiangyang 441053, China; liudezheng@hbuas.edu.cn (D.L.); liyan@hbuas.edu.cn (Y.L.); wenli_zhu@hbuas.edu.cn (W.Z.)

[2] ARC Centre of Excellence for Electromaterials Science, Intelligent Polymer Research Institute, Australia Institute of Innovative Materials (AIIM), Innovation Campus, University of Wollongong, Wollongong 2500, Australia

\* Correspondence: zhaojing@hbuas.edu.cn (J.Z.); liangxu@uow.edu.au (L.L.)

**Abstract:** The FeCoCrNiB$_x$ high-entropy alloy (HEA) coatings with three different boron (B) contents were synthesized on Q245R steel (American grade: SA515 Gr60) by laser cladding deposition technology. Effects of B content on the microstructure and wear properties of FeCoCrNiB$_x$ HEA coating were investigated. In this study, the phase composition, microstructure, micro-hardness, and wear resistance (rolling friction) were investigated by X-ray diffraction (XRD), a scanning electron microscope (SEM), a micro hardness tester, and a roller friction wear tester, respectively. The FeCoCrNiB$_x$ coatings exhibited a typical dendritic and interdendritic structure, and the microstructure was refined with the increase of B content. Additionally, the coatings were found to be a simple face-centered cubic (FCC) solid solution with borides. In terms of mechanical properties, the hardness and wear resistance ability of the coating can be enhanced with the increase of B content, and the maximum hardness value of three HEA coatings reached around 1025 HV$_{0.2}$, which is higher than the hardness of the substrate material. It is suggested that the present fabricated HEA coatings possess potentials in application of wear resistance structures for Q245R steel.

**Keywords:** high-entropy alloy; coating; laser cladding; microstructure; wear resistance

## 1. Introduction

Recently, high-entropy alloys (HEAs), which are defined as solid solution alloys that contain more than five principal elements in equal or near equal atomic percent (at.%) [1], have drawn rising interest from the materials science and engineering community since the first few papers on the subject were published in 2004 [2,3]. Due to the high-entropy effect in thermodynamics and hysteresis diffusion effect in dynamics [4,5], HEAs are usually composed of single solid solution phases, such as face-centered cubic (FCC) or body-centered cubic (BCC) structures, rather than complex intermetallic compounds. These particular structures with proper composition may contribute to the advantages of HEAs in such aspects as high mechanical strength, good ductility, high wear resistance, good resistance to oxidation and corrosion, etc. [6–8], providing more possible engineering applications in various fields.

To date, the fabrication routes of HEAs are mainly focused on the HEA bulk ingots and the HEA coatings [9]. The HEA bulk ingots are usually fabricated by the arc melting technique [10] or the casting method [11]; however, these techniques have a limited size of ingot due to the formation of the simple solid solution phase in the HEAs, which requires a high cooling rate [9]. Furthermore, given that HEAs comprise multiple expensive elements in high content, it is costly to fabricate the

HEA bulk ingots [12]. Thus, it is attractive to fabricate the HEA coatings with excellent properties on the cost-effective steel substrates. The alternative laser cladding deposition technique is used to fuse a designed alloy coating with about 1–5 mm thickness on the surface of a low cost iron substrate with a rapid solidification rate ($10^4$–$10^6$ °C/s), which leads to significant effects of non-equilibrium solute trapping, avoiding component segregation and improving solubility of the coating [13,14].

Therefore, significant efforts have been made to investigate the microstructure and mechanical properties of HEA coating by laser cladding [15–18]. For example, Zhang et al. [15] investigated the influences of silicon (Si) (1.2 mol.%), manganese (Mn) (1.2 mol.%), and molybdenum (Mo) (2.8 mol.%) additions on the microstructure, properties, and coating quality of laser-clad FeCoNiCrCu high-entropy alloy coating and found that the FeCoNiCrCu coatings with or without Si, Mn, and Mo additions are both identified to be simple FCC solid solutions. Besides, the micro-hardness is much higher than that of the alloy prepared by the arc melting technique with the same composition. Ye et al. [16] studied the microstructure of the laser cladding $Al_x$FeCoNiCuCr coating, the effects of aluminum (Al) element content on the coating hardness, and the high-temperature micro-hardness of the coating. Chuang et al. [17] reported that the strengthening methods for HEA coating can be performed through substitutional solid solution strengthening, by the addition of elements with large atomic radii, such as Al and titanium (Ti), to induce high lattice distortion. However, Yang et al. [18] indicated that the increased lattice strain and defects often lead to high brittleness and reduce the ductility and toughness of HEA coating. Recently, some studies [19–22] reported the addition of small elements, such as boron (B) or carbon (C), can be used to improve the mechanical properties of HEAs. Zhang et al. [23] indicated that it is vital to improve the solid solubility of B and control the boride morphology through the addition of the B element in the fabrication of new HEAs. However, little research has been published on the effects of B content on the microstructure and wear properties of the laser alloyed HEA coatings. According to Archard's law [24], the wear resistance of materials is proportional to the hardness. Furthermore, the wear resistance and potential industrial application value of FeCoCrNi HEAs (B free alloys) were reported by [25,26], and the FeCoCrNi HEAs presented low Vickers hardness and yield strength (141 HV and 145 MPa, respectively) [26]. Thus, the main purpose of this work is to control the B content in laser-clad $FeCoCrNiB_x$ (*x*: molar ratio, *x* = 0.5, 1, and 1.5, denoted by $B_{0.5}$, $B_{1.0}$, and $B_{1.5}$, respectively) HEA coatings and to explore the effects of B content on the microstructure and wear properties of these HEA coatings.

In this work, we prepared $FeCoCrNiB_x$ HEA coatings by laser cladding on a low carbon steel substrate (Q245R steel). The phase composition, microstructure, micro-hardness, and wear resistance (rolling friction) of the HEA coatings with different B content by laser cladding were investigated by X-ray diffraction (XRD), a scanning electron microscope (SEM), a micro hardness tester, and a roller friction wear tester, respectively. The significance of this research is the effects of B content on the microstructure and wear properties of laser-clad $FeCoCrNiB_x$ HEA coatings. The results suggest that the present fabricated HEA coatings possess potential in the application of wear resistance structures for low carbon steels.

## 2. Materials and Methods

Q245R steel plates (American grade: SA515 Gr60), which were provided by Baowu Steel Company (Wuhan, China), were used as the substrate material. Q245R steel plates were cut into rectangular specimens of dimensions 120 mm × 60mm × 10 mm by a wire cutting machine. The sizes of the substrate plates are shown in Figure 1. The chemical composition was measured using the PDA-5500S Shimadzu optical emissions spectrometers (Shimadzu Corporation, Kyoto, Japan). The Q245R substrate plate surface was treated by abrasive papers and then dried after using alcohol to remove the dirt and oil. A comparison of measured chemical composition of the Q245R substrate plate and the standard of GB713-2014 (national standard of the people's republic of China for alloy structural steel) for Q245R steel is presented in Table 1. It can be seen in Table 1 that the chemical composition of the specimen used in this study meets the requirement of the national standard.

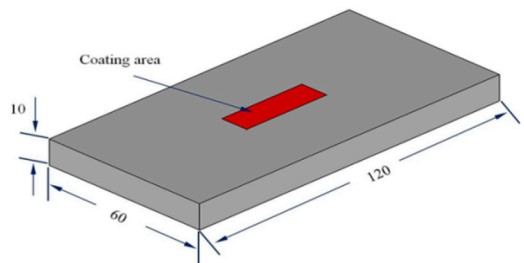

**Figure 1.** Schematic diagram of specimen sizes (mm).

**Table 1.** A comparison of the measured chemical composition of the Q245R substrate plate and the standard of GB713-2014 for Q245R steel (wt.%).

| wt.% | C | S | P | Mn | Si | Alt |
|---|---|---|---|---|---|---|
| Q245R | 0.18 | 0.012 | 0.022 | 0.65 | 0.022 | 0.031 |
| GB/T3077-2015 | ≤0.20 | ≤0.015 | ≤0.025 | 0.50–1.0 | ≤0.35 | ≥0.020 |

S: sulfur, P: phosphorus, Alt: aluminum total.

Alloy power mixtures with a nominal composition of $FeCoCrNiB_x$ ($x$: molar ratio, $x = 0.5$, 1, and 1.5, denoted by $B_{0.5}$, $B_{1.0}$, and $B_{1.5}$, respectively) were prepared by mechanically mixing commercially pure elemental powders (99.99 wt.%) with particle sizes below 400 mesh. The mixed powders were put into a vacuum oven at 80 °C to dry for 1 h after vacuum ball milling in order to remove moisture and enhance the liquidity of the powder. The plasma spraying technology [27] was adopted in the preparation of the $FeCoCrNiB_x$ HEA coatings. An appropriate amount of alcohol was taken as the adhesive and mixed with the aforementioned ball milling powder to obtain the viscous powder, which was coated on the Q245R steel matrix with a coating size of 32 mm × 10 mm × 1.6 mm. The coating was then put into the vacuum oven at 120 °C to dry for 2 h. Subsequently, Laser cladding was carried out using the YLS-4000-S2T-CL fiber laser system (IPG Photonics Corporation, Oxford, USA). After a series of optimization runs, the processing parameters were as follows: The laser radiation was at 980 nm wavelength, 4000 W output power, 1.5 mm × 12 mm rectangular beam, and 180 mm/min scanning speed velocity. High-purity argon gas was used as shielding gas through the coaxial nozzle to prevent oxidation. Figure 2 presents the macroscopic morphology of $FeCoCrNiB_x$ HEA coatings, and the three different B contents were 0.5, 1.0, and 1.5, respectively.

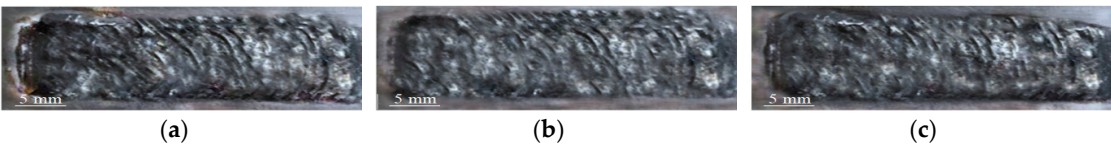

(**a**)            (**b**)            (**c**)

**Figure 2.** The macroscopic morphology of $FeCoCrNiB_x$ coatings with the three different molar ratios of B. (**a**) $FeCoCrNiB_{0.5}$, (**b**) $FeCoCrNiB_{1.0}$, and (**c**) $FeCoCrNiB_{1.5}$.

The crystal structures of the three different thickness HEA coatings were identified on the cross-section surfaces, using Bruker D8 Advance X-ray diffraction equipment AXS (Bruker AXS GMBH, Karlsruhe, Germany). The working voltage and current were 30 kV and 15 mA, respectively. The scan range was from 20° to 100°, and the scanning rate was 4°/min. For microstructural observation, the microstructures and chemical compositions of the HEA coatings were analyzed with the use of an S-4800 scanning electron microscope (Hitachi, Tokyo, Japan) and energy dispersive spectroscopy (EDS). The Vickers micro-hardness of the polished longitudinal-section surfaces of the HEA coatings was measured using an HMV-G21ST Vickers hardness tester (Shimadzu Corporation, Kyoto, Japan), and the test parameter was applied for 5 s under a load of 100 g. Each longitudinal-section surface was tested for multi points from the surface to the substrate with an equal interval. The wear tests were carried out using the M-2000A roller friction wear tester (YNSJ Test Instrument co., LTD, Jinan China)

at room temperature with a rotation speed of 180 r/min, a test load of 300 N, and a duration time of 20 min, and the counter body of roller was a W18Cr5V steel. The wear mass loss was measured by a precision balance instrument through weighing the samples before and after wear tests.

## 3. Results and Discussions

### 3.1. Microstructure, Phase Formation, and Phase Composition

The composition of the FeCoCrNiB$_x$ ($x$ = 0.5, 1.0, and 1.5, respectively) mixed powders before the laser cladding, determined by the EDS detector, is shown in Table 2. Although it is difficult to accurately measure the ratio of light element B by the EDS detector (5–10% error of our EDS instrument is considered), the trend of the B ratio can be well specified by EDS. In order to improve the measurement accuracy, the EDS result was collected from many data, and the average ratio was applied to eliminate the error in this study. It can be seen from Table 2 that the composition uniformity was ensured and there was no apparent macroscopic milling loss to the elements during the process of vacuum ball milling. Figure 3 presents the interface microstructure of the polished longitudinal sections of FeCoCrNiB$_x$ HEA coatings. As can been seen in Figure 3, most of the planar grains were concentrated at the interface between the HEA coating and substrate, and a few of the columnar grains appeared in the middle of the HEA coating.

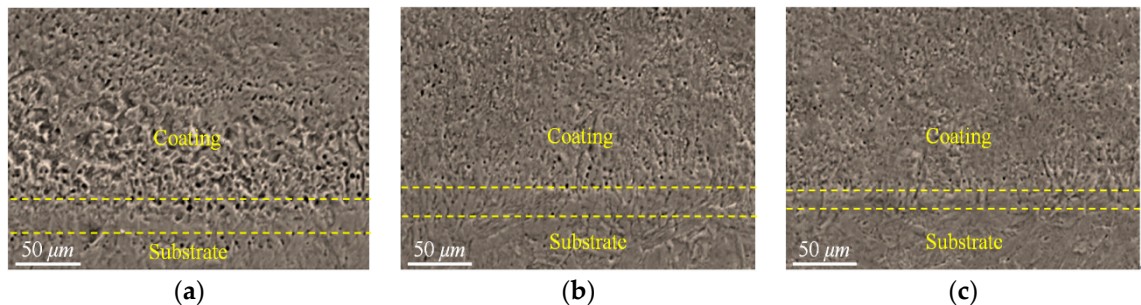

| (a) | (b) | (c) |

**Figure 3.** Interface microstructure of FeCoCrNiB$_x$ laser high entropy alloying layer. (**a**) FeCoCrNiB$_{0.5}$, (**b**) FeCoCrNiB$_{1.0}$, and (**c**) FeCoCrNiB$_{1.5}$.

**Table 2.** Chemical compositions (in at.%) of FeCoCrNiB$_x$ ($x$ = 0.5, 1.0, and 1.5, respectively) mixed powders.

| Alloy Type/Element | Fe | Co | Cr | Ni | B |
|---|---|---|---|---|---|
| FeCoCrNiB$_{0.5}$ | 22.64 | 21.92 | 21.95 | 22.36 | 11.13 |
| FeCoCrNiB$_{1.0}$ | 19.62 | 19.88 | 20.35 | 19.63 | 20.52 |
| FeCoCrNiB$_{1.5}$ | 18.37 | 17.95 | 18.13 | 18.32 | 27.23 |

It can be seen from Figure 3a that the coating and substrate were poorly connected and a local loosening phenomenon was observed. As shown in Figure 3b,c, a better metallurgical bonding between the coating and substrate was observed with the increase of B content. This phenomenon was ascribed to the constant increase in B content, which resulted in the increased degree of segregation of the elements in the HEA coating, thereby reducing the solidification temperature of frontier liquids and resulting in easier crystallization. Rapid cooling of alloy elements during the cladding process can increase the nucleation rate markedly and can eventually refine the microstructure of the HEA coating.

Figure 4 presents the XRD patterns of the laser-clad FeCoCrNiB$_x$ ($x$ = 0.5, 1.0, 1.5) HEA coatings. It was noticed that, in Figure 4a, all coatings displayed a mixture of the major face-centered cubic structure (FCC) solid solution matrix and the secondary boride precipitations. The main boride was the M$_2$B phase (M refers to Fe, Co, Cr, and Ni). According to Figure 4a, the phases in the FeCoCrNiB$_{0.5}$ coating could be identified as a simple FCC solid solution with diffraction peaks at about $2\theta = 35.5°$, $51.0°$, $65.5°$, $81.0°$, and $96.0°$ and a very small quantity of the M$_2$B phase with a diffraction peak at

about 57.5°. It also can be seen that the diffraction peaks corresponding to the FCC solid solution of FeCoCrNiB$_{1.0}$ were slightly weaker than that of FeCoCrNiB$_{0.5}$, whereas the peak at 57.5° was slightly enhanced, and a new diffraction peak corresponding to the M$_2$B phase was observed. For FeCoCrNiB$_{1.5}$ coating, some new phases have been identified as the boride precipitations at about 2θ = 28.5°, 48.5°, and 87°. This demonstrates that the constituent phase of coating changed from the FCC solid solution to a combination of the FCC solid solution and the M$_2$B phase with the increase of B content.

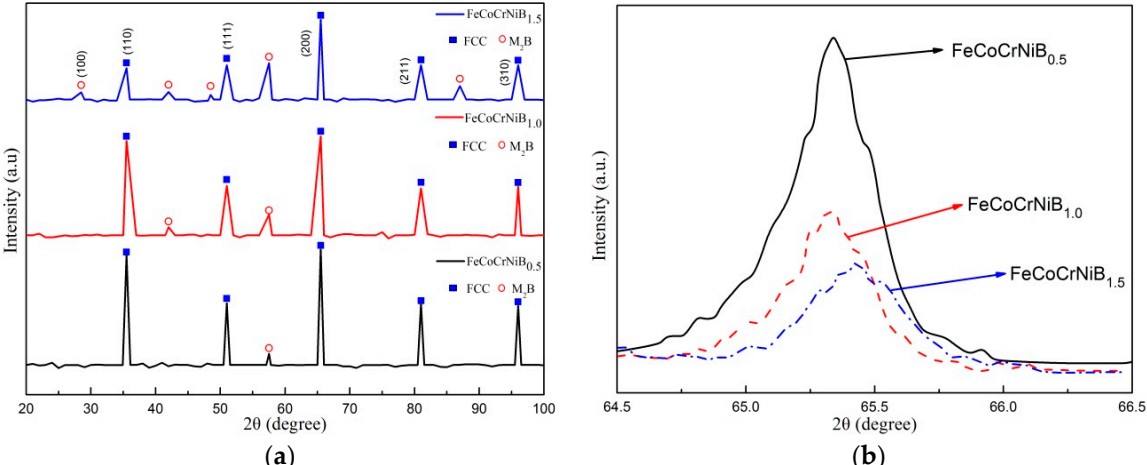

**Figure 4.** X-ray diffraction patterns of the laser-clad FeCoCrNiB$_x$ (*x* = 0.5, 1.0, and 1.5, respectively) high-entropy alloy (HEA) coatings. (**a**) 2θ from 20° to 100°, (**b**) 2θ from 64.5° to 66.5°.

Figure 4b shows an enlarged view of the diffraction peak of the FCC solid solution (211) in the FeCoCrNiB$_x$ HEA coating. Increasing the content of B from 0.5 to 1.5 can decrease the intensity of the diffraction peak corresponding to the FCC solid solution (211) and the shift of the diffraction peak to the higher angle. This phenomenon was due to the increasing B content, which can increase the number of soluble atoms of B in the crystal structure. However, the B atom was greatly different in terms of size, from the other atoms, thereby increasing lattice distortion. Thus, the diffuse reflection effect was strong, and the intensity of the diffraction peak was reduced. Moreover, the radius of the B atom was smaller than that of the other atoms, and its displacement ability to other elements was strong. With increasing displacement of the B atom to the other atoms in the crystal structure, the lattice constant decreased, thereby shifting the diffraction peak to the higher angle.

SEM images of the microstructures of the laser-clad FeCoCrNiB$_x$ HEA coatings with three different B contents are shown in Figure 5. The coatings exhibited a typical dendritic and interdendritic structure, and the microstructure was refined with the increase of B content. XRD analysis and the microstructure of FeCoCrNiB$_{0.5}$ coating in Figure 5a shows that the dendrites were identified to belong to the primary FCC solid solution and interdendrites were identified to belong to the FCC solid solution phase and a small quantity of the M$_2$B phase. According to Figure 5a–c, the volume fraction of the M$_2$B phase increased and dendrites decreased gradually with the increase of the molar ratio of B content. The FeCoCrNiB$_{1.0}$ coating was composed of the eutectic structure of the FCC solid solution and the M$_2$B phase. When the molar ratio of B content increased from 1.0 to 1.5, the coating was composed of the eutectics of willow-like primary M$_2$B phase, M$_2$B phase, and FCC solid solution phase, and the microstructure tended to blocky.

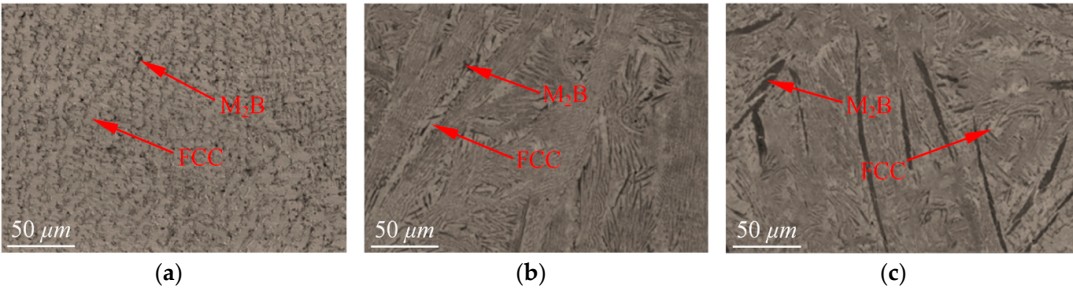

**Figure 5.** Scanning electron microscope (SEM) images of the microstructures of the laser-clad FeCoCrNiB$_x$ HEA coatings. (**a**) FeCoCrNiB$_{0.5}$, (**b**) FeCoCrNiB$_{1.0}$, and (**c**) FeCoCrNiB$_{1.5}$.

For mapping the compositional distribution, EDS map analysis of the alloy elements are presented in Figure 6. The qualitative segregation characteristics can be visualized, and the EDS results of FeCoCrNiB$_x$ coatings are shown in Table 3. According to Figure 6a–f, when the molar ratio of B content was 0.5, Fe, Co, and Ni were increased in terms of the amount in the gray dendritic regions, while Cr and B were reduced in the dendritic structure. XRD analysis showed that the main gray dendrite structure was the FCC solid solution. The Fe content in the white interdendritic structure was the highest, followed by Cr and B, thereby indicating slight component segregation. A small number of borides was also observed in the structure. According to the XRD results, the gray dendritic and white interdendritic structures were the FCC solid solution. Increasing the molar ratio of B content to 1.0 resulted in a complex coating structure. The EDS results in Figure 6g–l show that the contents of Cr and B in the dendritic structure were the lowest, whereas those of Fe, Co, and Ni were higher. The XRD analysis showed that the gray dendritic structure was composed of the FCC solid solution, whereas the white interdendritic structure was composed of the FCC solid solution and a small amount of the M$_2$B phase. As shown by the EDS results in Figure 6m–r, when the molar ratio of B content was increased to 1.5, large amounts of Fe, Co, and Ni were enriched in the gray dendritic structure. The XRD results showed that the gray dendritic structure was composed of the FCC solid solution, whereas the white interdendritic structure was composed of the eutectic structure of the FCC solid solution and the M$_2$B phase. In the long strip structure of the M$_2$B phase, a large amount of B was enriched, followed by Cr and F. Moreover, the atomic ratio of Cr to B was about 2:1. With the increase of the content of B, the M$_2$B phase separated from the intergranular structure. The M$_2$B phase was mainly composed of Fe and Cr borides, and the Cr from the interdendritic structure was segregated gradually to the M$_2$B phase.

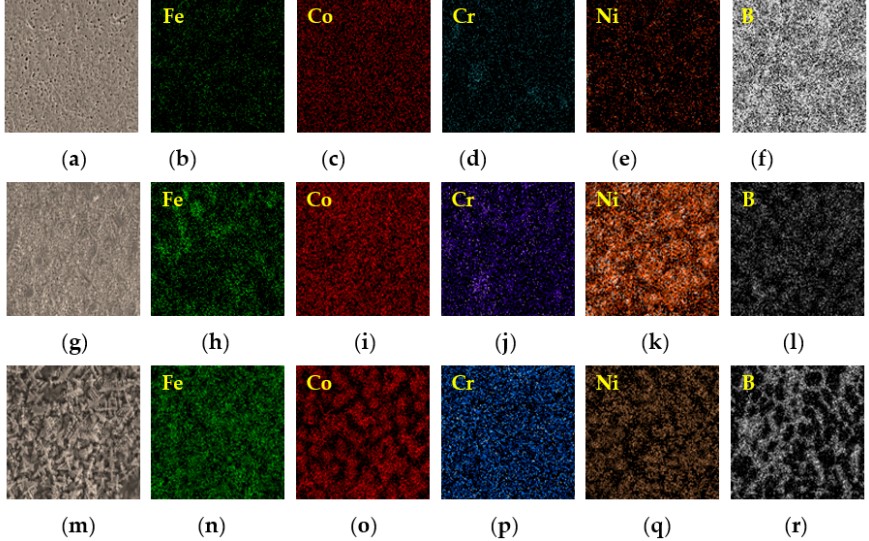

**Figure 6.** Energy dispersive spectroscopy (EDS) maps of the laser-clad FeCoCrNiB$_x$ HEA coatings. (**a**–**f**) FeCoCrNiB$_{0.5}$, (**g**–**l**) FeCoCrNiB$_{1.0}$, and (**m**–**r**) FeCoCrNiB$_{1.5}$.

**Table 3.** EDS results of FeCoCrNiB$_x$ HEA coatings (in at.%).

| Coating Type | Zone | Fe | Co | Cr | Ni | B |
|---|---|---|---|---|---|---|
| FeCoCrNiB$_{0.5}$ | Nominal | 22 | 22 | 22 | 22 | 12 |
| | FCC | 26 | 24 | 19 | 21 | 10 |
| | M$_2$B | 27 | 16 | 30 | 11 | 16 |
| FeCoCrNiB$_{1.0}$ | Nominal | 20 | 20 | 20 | 20 | 20 |
| | FCC | 22 | 19 | 10 | 24 | 25 |
| | M$_2$B | 22 | 18 | 22 | 8 | 30 |
| FeCoCrNiB$_{1.5}$ | Nominal | 18 | 18 | 18 | 18 | 28 |
| | FCC | 21 | 18 | 11 | 35 | 15 |
| | M$_2$B | 22 | 9 | 26 | 6 | 37 |

Yeh et al. [3] proposed that HEAs should contain at least five elements and that the entropy of mixing is the main factor that promotes the formation of a multi-component solid. The largest atomic radius difference among the Fe, Co, Cr, Ni, and B elements could be calculated by the following formula [28]:

$$\Delta R_{max} = \max(R_i - R_a)/R_a, \tag{1}$$

where $R_a$ is the average atomic radius and $R_i$ represents the atomic radius of any element. According to the formula of the Gibbs free energy [29]:

$$\Delta G_{min} = \Delta H_{mix} - T\Delta S_{mix}, \tag{2}$$

where $\Delta G_{mix}$ is the change in Gibbs free energy before and after the phase change, $\Delta H_{mix}$ is the mixing enthalpy, $\Delta S_{mix}$ is the mixing entropy, and $T$ is the absolute temperature. The mixing entropy $\Delta S_{mix}$ can be calculated by the following formula:

$$\Delta S_{mix} = -R\sum_{i=1}^{n} C_i ln C_i \tag{3}$$

where $R$ is constant gases and $R = 8.31\text{J}\cdot\text{K}^{-1}\cdot\text{mol}^{-1}$ and $C_i$ is the molar fraction of each element [29].

According to Equation (2), the high $\Delta S_{mix}$ can significantly lower the free energy of solid solution with multi principal elements, thus lowering the tendency to order and segregate, which consequently allows the solid solution to more easily form and be more stable than intermetallics or other ordered phases during the solidification of alloys [15]. However, the kinetic effects by laser rapid solidification also play a more important role on the phase formation in the coating. Lin et al. [30] indicated that not all HEAs only form solid solution phases during solidification, and small amount of compound phases are also separated from some HEA precipitates. Based on Equation (2), whether the alloy solidifies into a simple solid solution or into a compound is determined commonly by the $\Delta H_{mix}$ and the $\Delta S_{mix}$. It is noted that Hsu et al. [31] reported that the FeCoCrNiB$_x$ may form the $\Delta H_{mix}$ of binary boride. As shown in Table 4, Fe and Cr borides had large $\Delta H_{mix}$, whereas Co and Ni borides had small $\Delta H_{mix}$. Therefore, the contents of Fe and Cr in the produced borides (M$_2$B) were high, whereas those of Co and Ni were relatively low. These results are consistent with those of the energy spectrum measurement.

**Table 4.** Enthalpies of binary borides that might be formed in the alloy.

| M$_2$B | Fe$_2$B | Co$_2$B | Cr$_2$B | Ni$_2$B |
|---|---|---|---|---|
| $\Delta H_{mix}$ (KJ/mol) | 30 | 28 | 34 | 27 |

### 3.2. Microhardness and Wear Properties

The Vickers micro-hardness of the different B mole HEAs was measured from the substrate to the metallurgical bonding surface and then the cladding layer. Each cross section was tested for

multi points from the substrate to the surface with an equal interval of 0.2 mm. Figure 7 shows the micro-hardness distribution ranging from the Q245R substrate to the surface of the coatings with various B additions. The average thickness of the coating was about 1.6 mm.

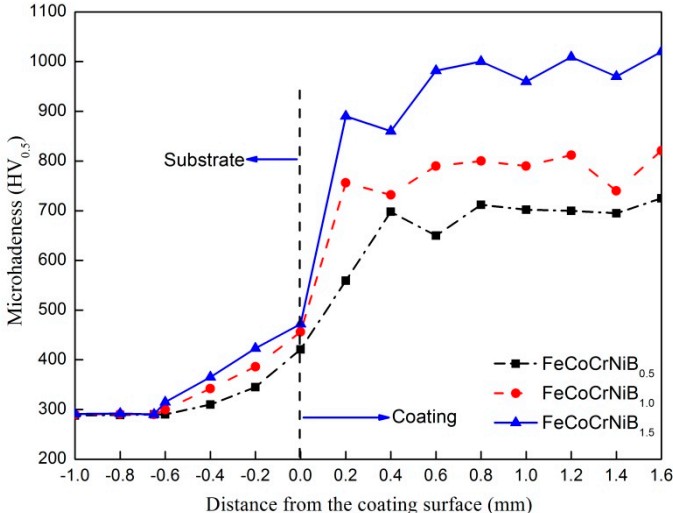

**Figure 7.** Micro-hardness profile along the cross section of the FeCoCrNiB$_x$ HEA coatings with various boron (B) additions.

It can be seen that the micro-hardness value of coating is much higher than that of the Q245R substrate material, and the maximum hardness value in three HEA coatings reached around 1025 HV$_{0.2}$ when the molar ratio of B was 1.5. With the increase of the cladding height, no obviously regular change was found in the micro-hardness. Moreover, with the increase of the content of B, the hardness showed an increasing tendency. The reasons for this can be attributed to the grain refinement, Nano-precipitates, and increased crystal packing density in the coating due to the rapid cooling rate in the laser cladding process. Meanwhile, based on the changes in the contents of borides in all coatings with the addition of B, the gradual increase in hardness was related to the increase of the hard phase of boride. The lattice distortion was caused by the solid solution of elements in the matrix, which increased the resistance of the dislocation motion and made the gliding of the dislocation difficult, thus improving the strength and hardness of the alloy [32]. Since the gap formed by the solid solution of the HEA is small, it is generally believed that the form of the atomic solid solution belongs to the displacement solid solution, so the lattice distortion is mainly caused by the difference of atomic size. In order to quantitatively analyze the influence of atomic size on lattice distortion, the standard deviation of atomic size between dendrites was calculated as follows:

$$a_{avg} = \sum\nolimits_{i=0}^{n} a_i c_i \tag{4}$$

$$s = \sqrt{\sum\nolimits_{i=0}^{n} \left(a_i - a_{avg}\right)^2 c_i} \tag{5}$$

where $a_i$ is the atomic size, $C_i$ refers to the molar fraction of each element, $a_{avg}$ is the average atomic size, and s is the standard deviation. The values of the atomic radius of Fe, Co, Cr, and Ni were very similar, which were 1.27, 1.26, 1.27, and 1.24, respectively. The standard deviation of Fe, Co, Cr, and Ni was very small. However, the value of the atomic radius of B was 0.95, which was much smaller than the other four principal components. As a result, the standard deviation was larger than FeCoCrNi. During the laser cladding process, a part of the B atom formed the hardness phase of borides with other metal atoms, and the other part of the B atom, as an interstitial atom, was dissolved in the FCC solid solution, i.e., dissolved in the matrix of the FCC solid solution. The rapid solidification conditions during laser cladding increased the limit of solid solubility of the atoms in the solid solution. Thus, more B atoms

were solidified in the solid solution. According to the XRD results, the addition of the element B can increase the lattice constant of the FCC solid solution. Thus, the increase of micro-hardness was caused by the increasing crystal packing density and lattice distortion of the FCC crystalline.

Figure 8 shows the wear morphology of FeCoCrNiB$_x$ HEA coating. It can be seen from Figure 8 that the wear volume of the coating decreased with the increase of the B ratio. Figure 9 shows the relationship between the wear volume and the average hardness of FeCoCrNiB$_x$ HEA coatings. The wear volume of the coating increased gradually with decreasing average hardness. The wear resistance of FeCoCrNiB$_x$ HEA coatings was positively correlated with the hardness. The wear resistance of FeCoCrNiB$_{1.5}$ HEA coating was the highest and the maximum hardness value of three HEA coatings was the FeCoCrNiB$_x$ HEA coating, and the hardness was around 1025 HV$_{0.2}$, whereas that of the FeCoCrNiB$_{0.5}$ HEA coating was the lowest. In this work, the wear resistance and hardness of the coating conformed to the classic Archard law [24], which states that the wear resistance of materials is proportional to the hardness.

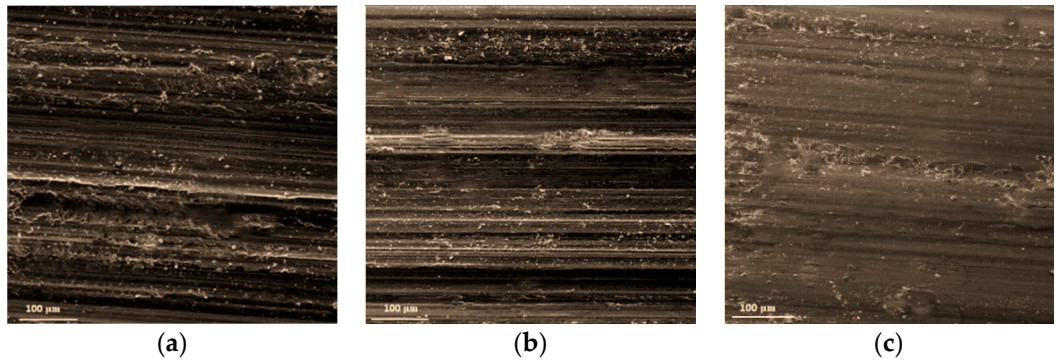

**Figure 8.** Wear morphology of the FeCoCrNiB$_x$ HEA coating. (**a**) FeCoCrNiB$_{0.5}$, (**b**) FeCoCrNiB$_{1.0}$, and (**c**) FeCoCrNiB$_{1.5}$.

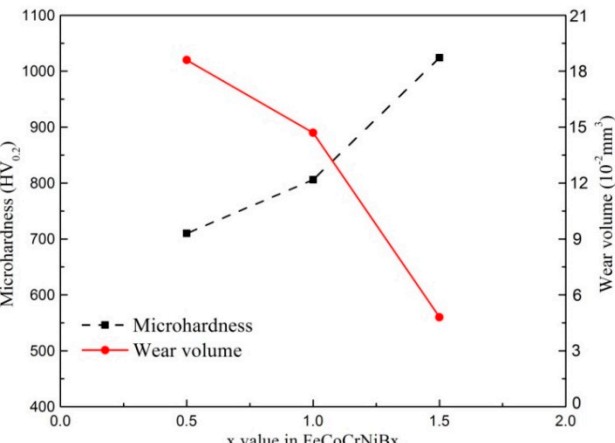

**Figure 9.** Micro-hardness profile along the cross-section.

## 4. Conclusions

In brief, the FeCoCrNiB$_x$ HEA coatings with three different boron contents were successfully synthesized on Q245R steel through the preset paving method by using laser cladding deposition technology. The microstructure of the FeCoCrNiB$_x$ HEA coatings exhibited a typical dendritic and interdendritic structure, and the coating was found to be a simple FCC solid solution with borides. After studying the effect of the content of B on the performance of FeCoCrNiB$_x$ coatings, we found that the microstructure of coatings could be refined with the increase of B content. Furthermore, the increase of B element facilitated the formation of the M$_2$B phase from the FCC matrix for the FeCoCrNiB$_x$ alloys and the formation of the M$_2$B phase could effectively enhance the wear properties

of the alloys. The relationship between the wear volume and the average hardness of FeCoCrNiB$_x$ HEA coatings is also reported. Additionally, the maximum hardness coating of three HEA coatings was the FeCoCrNiB$_x$ HEA coating, and the hardness value reached around 1025 HV$_{0.2}$, which was higher than the hardness of the substrate material. This work indicates that the present fabricated HEA coatings possess the potentials in the application of wear resistance structures for Q245R steel.

**Author Contributions:** Conceptualization, D.L.; methodology, D.L. and J.Z.; software, D.L. and Y.L.; validation, Y.L.; formal analysis, W.Z.; writing—original draft preparation, D.L.; writing—review and editing, J.Z. and L.L.; visualization, Y.L.; supervision, L.L.; project administration, D.L. All authors have read and agreed to the published version of the manuscript.

**Funding:** This research was funded by the Hubei Superior and Distinctive Discipline Group of "Mechatronics and Automobiles" (No. XKQ2019009).

**Conflicts of Interest:** The authors declare no conflict of interest.

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
