# Peer review of "Effects of Boron Content on Microstructure and Wear Properties of FeCoCrNiBx High-Entropy Alloy Coating by Laser Cladding"

_applsci, doi:10.3390/app10010049_

Round 1

Reviewer 1 Report

In this work, the authors prepared FeCoCrNiBx HEA coatings by laser cladding on a low carbon steel substrate (Q245R steel). The phase composition, microstructure, micro-hardness and wear resistance (rolling friction) of the HEA coatings with different B content by laser cladding were investigated by the X-ray diffraction (XRD), scanning electron microscope (SEM), micro hardness tester and roller friction wear tester, respectively. The significance of this research is that the effects of B content on the microstructure and wear properties of laser-clad FeCoCrNiBx HEA coatings. The result suggested that the present fabricated HEA coatings possessed the potentials in application of wear resistance structures for the low carbon steels. In my opinion this work is interesting and is publishable as it is .

Author Response

Thank you very much for your work concerning my paper. We are grateful to your comments for the manuscript. We hope our paper can be finally acceptable for publication in Applied Sciences (ISSN 2076-3417). Thank you very much for your recognition on my paper.

Wish you all the best!

Sincerely yours,

Dezheng Liu

Reviewer 2 Report

In the opinion of the reviewer the article may be published in this initial version.

Author Response

(The authors gave the same response as above.)

Reviewer 3 Report

The paper reports some experimental results on laser cladding process aimed at producing high entropy alloys deposited on a substrate.

The work is pretty nice and the interest on these allloys is growing nowadays.

However, some parts needs to be improved:

to study the effect on B content should be better to investigated the alloy without B, so the authors are invited to reports the characteristic of the cladded parts with the B free alloy I have some doubts about the correct measurement of B, due to the fact that EDS is not well calibrated for this element table 2 reports the EDS measurements of the claddings, however the EDS compositions of the initial powder shoudl be reported too. This allows to affirm if there is or not compositional variation during the laser cladding table 3: two numbers after coma are not significant values for EDS. please, report just the integer number figure 7: there is no constant HV in the base material, so that means that the authors are invited to proceed with the HV profiles deeper in the substrate in order to find the real HV of the base material figure 8 regards the functional characterization of the cladded parts. It is very important to link these data with the SEM images of the surfaces before and after the wearing testing to observe the type of material behaviour. Even here, the comparison with the cladding with the B free alloy makes sense to justify the needs of B for enhancing the tribological response 

Author Response

Dear reviewer:

I am very grateful to your comments for the manuscript. We also highly appreciate your carefulness, conscientious, and the broad knowledge on the relevant research fields, since you have given us a number of beneficial suggestions. According to the comments we received, we have made the following revisions on this manuscript (Please see the attachment for details):

Point 1: The paper reports some experimental results on laser cladding process aimed at producing high entropy alloys deposited on a substrate. The work is pretty nice and the interest on these alloys is growing nowadays. However, some parts needs to be improved: to study the effect on B content should be better to investigated the alloy without B, so the authors are invited to reports the characteristic of the cladded parts with the B free alloy I have some doubts about the correct measurement of B, due to the fact that EDS is not well calibrated for this element table 2 reports the EDS measurements of the claddings, however the EDS compositions of the initial powder should be reported too. This allows to affirm if there is or not compositional variation during the laser cladding.

Response 1: Thank you for your beneficial suggestions. A revised manuscript with the correction sections red marked (with Track changes) was attached as the supplemental material and for easy check or editing purpose.

In previous studies, the mechanical properties and potential industrial application value of FeCoCrNi-based alloys were widely investigated such as [1] J. Alloys. Compd. 2017, 695, 1479-1487 and [2] Acta Mater. 2016, 116, 332-342. Furthermore, the hardness and wear resistance of FeCoCrNi HEAs (B free alloys) were reported by [3] Mater. Chem. Phys. 2017, S025405841730425X., and the FeCoCrNi HEAs exhibited low Vickers hardness and yield strength (141 HV and 145 MPa, respectively). And many other alloy elements, such as Ta, Al, Mo, Ti, V, Nb, and Si, were added into FeCoCrNi alloy, aimed to improve the mechanical properties of HEAs and the relevant research results have been published in: ([4] J. Alloys Compd. 591 (2014) 11-21.; [5] Intermetallics. 60 (2015) 1-8.; [6] J. Alloys Compd. 656 (2016) 284-289.; [7]  Mater. Sci. Eng. A 648 (2015) 15-22.; [8] Mater. Charact. 70 (2012) 63-67.; [9] Intermetallics 44 (2014) 37-43.) .The poor wear resistance of FeCoCrNi HEAs (B free alloys) has been verified by many researchers, thus we only designed the B content in laser-clad FeCoCrNiBx (x: molar ratio, x = 0.5, 1 and 1.5 denoted by B0.5, B1.0 and B1.5, respectively) HEA coatings in order to explore the effects of B content on the microstructure and wear properties of these HEA coatings. The demonstrations for why we investigated the alloy without B have been added in the revised version. Details see: Line 71-75 in the revised version. 

We have carefully checked the description for the measurement of B. EDS is indeed not well calibrated for the light element such as B and N. Although it is difficult to accurately measure the ratio of light element B by EDS detector (5-10 % error of our EDS instrument is considered), the trend of the B ratio can be well specified by EDS. In order to improve the measurement accuracy, our EDS result was collected from many data and the average ratio was applied to eliminate the error in this study. Details see: Line 140-145 in the revised version. (Although it is difficult to accurately measure the ratio of light element B to eliminate the error in this study).

Furthermore, EDS technology has been adopted in the research of B-containing alloys and has relatively high reliability. (For example: [1] Mater Design, 2012, 34, 637; [2] Mater Design, 2017, 127, 97-105)

Lastly, chemical compositions (in at.%) of FeCoCrNiBx (x = 0.5, 1.0 and 1.5, respectively) mixed powders before the laser cladding have been listed in Table 2 in order to ensure there is no apparent macroscopic milling loss to the elements during the process of vacuum ball milling. Details see: Line 151 -152 in the revised version.

Point 2: table 3: two numbers after coma are not significant values for EDS. please, report just the integer number.

Response 2: Thank you for your beneficial suggestions. Two numbers after coma are indeed not significant values for EDS. Thus, we have revised the table 3 and report just the integer number. Details see: Line 237 in the revised version.

Table 3. EDS results of FeCoCrNiBx HEA coatings (in at.%)

Coating type

Zone

Fe

Co

Cr

Ni

B

FeCoCrNiB0.5

Nominal

22

22

22

22

12

FCC

26

24

19

21

10

M2B

27

16

30

11

16

FeCoCrNiB1.0

Nominal

20

20

20

20

20

FCC

22

19

10

24

25

M2B

22

18

22

8

30

FeCoCrNiB1.5

Nominal

18

18

18

18

28

FCC

21

18

11

35

15

M2B

22

9

26

6

37

Point 3:  figure 7: there is no constant HV in the base material, so that means that the authors are invited to proceed with the HV profiles deeper in the substrate in order to find the real HV of the base material.

Response 3: Thank you for your beneficial suggestion. HV profiles deeper in the substrate have been presented in revised version. Details see: Line 268 -270 in the revised version. (Figure 7. Micro-hardness profile along the cross section of the FeCoCrNiBx HEA coatings with various B additions).

Figure 7. Micro-hardness profile along the cross section of the FeCoCrNiBx HEA coatings with various B additions.

Point 4: figure 8 regards the functional characterization of the cladded parts. It is very important to link these data with the SEM images of the surfaces before and after the wearing testing to observe the type of material behaviour. Even here, the comparison with the cladding with the B free alloy makes sense to justify the needs of B for enhancing the tribological response.

Response 4: Thank you for your good suggestion.  Wear morphology of FeCoCrNiBx HEA coating has been added in the revised version.  Details see Line 300-304 (Figure 8. Wear morphology of FeCoCrNiBx HEA coating. (a) FeCoCrNiB0.5, (b) FeCoCrNiB1.0 and (c) FeCoCrNiB1.5.) Figure 8 shows the wear morphology of FeCoCrNiBx HEA coating. It can be seen from Figure 8 that the wear volume of the coating decreased with the increase of the B ratio. Figure 9 shows the relationship between the wear volume and the average hardness of FeCoCrNiBx HEA coatings. The significance of this research is that the effects of B content on the microstructure and wear properties of laser-clad FeCoCrNiBx HEA coatings. Due to the samples before the wearing testing have been exhausted and the fiber laser system in our laboratory has been in a state of upgrade and maintenance, it is difficult to prepared FeCoCrNi now. Furthermore, the hardness and wear resistance of FeCoCrNi HEAs (B free alloys) were reported by [3] Mater. Chem. Phys. 2017, S025405841730425X., and the FeCoCrNi HEAs exhibited low Vickers hardness and yield strength (141 HV and 145 MPa, respectively). And many other alloy elements, such as Ta, Al, Mo, Ti, V, Nb, and Si, were added into FeCoCrNi alloy, aimed to improve the mechanical properties of HEAs and the relevant research results have been published in: ([4] J. Alloys Compd. 591 (2014) 11-21.; [5] Intermetallics. 60 (2015) 1-8.; [6] J. Alloys Compd. 656 (2016) 284-289.; [7]  Mater. Sci. Eng. A 648 (2015) 15-22.; [8] Mater. Charact. 70 (2012) 63-67.; [9] Intermetallics 44 (2014) 37-43.) .The poor wear resistance of FeCoCrNi HEAs (B free alloys) has been verified by many researchers, thus we only designed the B content in laser-clad FeCoCrNiBx (x: molar ratio, x = 0.5, 1 and 1.5 denoted by B0.5, B1.0 and B1.5, respectively) HEA coatings in order to explore the effects of B content on the microstructure and wear properties of these HEA coatings.

The grammatical or typographical errors have been revised. A revised manuscript with the correction sections red marked (with Track changes) was attached as the supplemental material and for easy check or editing purpose. Thank you very much for giving me an opportunity to revise our manuscript. We hope that these revisions are satisfactory and that the revised version will be acceptable for publication in Applied Sciences. Thank you very much for your work concerning my paper. Kind consideration of this manuscript is highly appreciated; however, if there are more questions, we are willing to revise it again. 

Wish you all the best!

Sincerely yours,

Dezheng Liu
